# A new Middle Jurassic diplodocoid suggests an earlier dispersal and diversification of sauropod dinosaurs

Xing Xu[1], Paul Upchurch[2], Philip D. Mannion [3], Paul M. Barrett [4], Omar R. Regalado-Fernandez [2], Jinyou Mo[5], Jinfu Ma[6] & Hongan Liu[7]

The fragmentation of the supercontinent Pangaea has been suggested to have had a profound impact on Mesozoic terrestrial vertebrate distributions. One current paradigm is that geographic isolation produced an endemic biota in East Asia during the Jurassic, while simultaneously preventing diplodocoid sauropod dinosaurs and several other tetrapod groups from reaching this region. Here we report the discovery of the earliest diplodocoid, and the first from East Asia, to our knowledge, based on fossil material comprising multiple individuals and most parts of the skeleton of an early Middle Jurassic dicraeosaurid. The new discovery challenges conventional biogeographical ideas, and suggests that dispersal into East Asia occurred much earlier than expected. Moreover, the age of this new taxon indicates that many advanced sauropod lineages originated at least 15 million years earlier than previously realised, achieving a global distribution while Pangaea was still a coherent landmass.

[1] Key Laboratory of Evolutionary Systematics of Vertebrates, Institute of Vertebrate Paleontology & Paleoanthropology, Chinese Academy of Sciences, 100044 Beijing, China. [2] Department of Earth Sciences, University College London, Gower Street, London WC1E 6BT, UK. [3] Department of Earth Science and Engineering, Imperial College London, South Kensington Campus, London SW7 2AZ, UK. [4] Department of Earth Sciences, Natural History Museum, Cromwell Road, London SW7 5BD, UK. [5] Natural History Museum of Guangxi, 530012 Nanning, Guangxi, China. [6] Lingwu National Geopark Administration, 750400 Lingwu, Ningxia, China. [7] Lingwu Historic Relic Administration, 750400 Lingwu, Ningxia, China. These authors contributed equally: Xing Xu, Paul Upchurch, Philip D. Mannion  Correspondence and requests for materials should be addressed to X.X. (email: xu.xing@ivpp.ac.cn)

S auropods were gigantic long-necked herbivorous dinosaurs that dominated many Jurassic and Cretaceous terrestrial faunas[1,2]. Although sauropods were globally distributed[2], several subgroups displayed restricted geographic ranges that potentially reflect endemism caused by the fragmentation of Pangaea[3]. The absence of Diplodocoidea from the otherwise rich sauropod faunas of East Asia has thus been interpreted as a genuine biogeographic pattern[4–6].

From 2005 onward, we organised four excavations at a new dinosaur site in the lower Middle Jurassic Yanan Formation at Ciyaopu, Lingwu, Ningxia Hui Autonomous Region, northwest China (Supplementary Notes 1 and 2, Supplementary Fig. 1), which resulted in the discovery of fossil material comprising 7–10 partial skeletons (including portions of two skulls), ranging from juveniles to adults. Here, we name a new sauropod based on this material, and demonstrate that it is, to our knowledge, the earliest diplodocoid (and therefore also the earliest neosauropod) sauropod, and the first from East Asia (Supplementary Notes 3–7). This discovery has major implications for calibrating the timing of neosauropod diversification, provides the first insight into a previously hidden aspect of their evolutionary history, and questions East Asia's hypothesized status as an isolated island continent during the Jurassic (Supplementary Notes 4–7).

## Results

### Systematic paleontology.

<div align="center">

Sauropoda Marsh, 1878
Neosauropoda Bonaparte, 1986
Diplodocoidea (Marsh, 1884)
Dicraeosauridae Janensch, 1929
*Lingwulong shenqi* gen. et sp. nov.

</div>

**Etymology**: *Lingwu*, after the region where the specimens were found; *long*, the Mandarin Chinese for 'dragon'; and *shenqi*, the Mandarin Chinese for 'amazing', reflecting the unexpected discovery of a dicraeosaurid in the Middle Jurassic of China.

**Holotype**: Lingwu Museum (LM) V001a, a partial skull comprising the braincase, skull roof, and occiput, and an associated set of dentary teeth (Fig. 1), found in quarry II.

**Paratype**: Lingwu Geopark (LGP) V001b. A semi-articulated partial skeleton including a series of posterior dorsal vertebrae, complete sacrum, the first caudal vertebra, partial pelvis, and incomplete right hind limb. The holotypic braincase was found close to the anterior end of the dorsal series and the right scapulocoracoid, suggesting that they potentially belong to the same individual. However, we conservatively treat them as separate individuals because the skull was not found in close association with cervical elements, and because there are other individuals nearby within the quarry.

**Referred specimens**: Institute of Vertebrate Paleontology and Paleoanthropology (IVPP) V23704, 29 dentary teeth preserved in a 'U'-shaped arc; LGP V002, a partial skeleton including several dorsal and caudal vertebrae, both scapulocoracoids, partial forelimbs, and partial pelvis; LGP V003, a partial skeleton including a nearly complete dorsal series and sacrum, two anterior caudal vertebrae, and both ilia; LGP V004, a small individual represented by an anterior cervical vertebra, an anterior dorsal vertebra, and right tibia; LGP V005, a partial skeleton preserving the sacrum, pelvis, and a semi-articulated series of 25 anterior and middle caudal vertebrae; LGP V006, a partial skeleton preserving several cervical vertebrae, incomplete scapulae and complete coracoids, and partial forelimb; and numerous disassociated elements.

**Horizon and locality**: Yanan Formation, late Early to early Middle Jurassic (late Toarcian–Bajocian), Lingwu Geopark, near Ciyaopu, Ningxia Hui Autonomous Region, China.

**Diagnosis**: Autapomorphies: prefrontal anterior process directed laterally; orbital dorsal margin strongly ornamented by deep, longitudinal grooves and tubercles; long-axes of the free tips of the basal tubera directed anteromedially; capitate process mediolaterally long (length:height ratio c. 5.0); occipital condyle articular surface wide transversely (width:height ratio c. 1.54); lateral surface of cervical prezygapophyseal process bears a ridge formed by a linear array of tubercles; subcircular facet-like region at the summit of metapophyses in middle cervical-anterior dorsal vertebrae; small process projects anterodorsally from the anterior margin of the transverse process, near its distal end, in anterior dorsal vertebrae; anterior dorsal metapophyses twisted along their length; anterior caudal neural spines bear subtriangular facet-like areas, extending from summit to spine mid-height.

## Description

Cranial material includes the skull roof, occiput, and braincase (Fig. 1, Supplementary Fig. 2). As in dicraeosaurids[7,8], the frontals are co-ossified along the midline. Supratemporal fenestrae are large and open dorsolaterally: this is the plesiomorphic state and contrasts with the reduced openings in many other dicraeosaurids and rebbachisaurids[7,8]. The frontoparietal suture is located midway between the anterior and posterior margins of the supratemporal fenestrae, as in advanced dicraeosaurids and some diplodocids[9]. There is a hook-like posteroventrally-directed process on the main body of the squamosal, as is also seen in a mild form in the diplodocid *Kaatedocus*, and prominently in dicraeosaurids[7–9]. In lateral view, the squamosal ventral process projects anteroventrally, implying the presence of an anteroventrally oriented quadrate and a long slit-like lower temporal opening, as is typical for diplodocoids[2,7]. The postorbital ventral process has a subtriangular transverse cross-section. This process is not strongly compressed anteroposteriorly, which is highly unusual for a eusauropod[1], though this morphology also occurs in the rebbachisaurid *Limaysaurus* (MUCPv-205). As in dicraeosaurids[7], the sagittal crest on the supraoccipital is prominent and plate-like in *Lingwulong*. *Suuwassea*[10], *Amargasaurus* (MACN N-15), and *Lingwulong* (but not *Dicraeosaurus* and other sauropods) possess a deep slot-like fossa on the occiput, lateral to the proatlantal facets (Fig. 1). Basal tubera are prominent, unlike the derived state in rebbachisaurids such as *Limaysaurus* and *Nigersaurus*, where they are very reduced[7]. Above the basipterygoid process, the left otosphenoidal ridge bears the broken base of a dorsoventrally flattened 'leaf'-like process, a derived state in dicraeosaurids[2,8]. In lateral view, with the skull roof horizontal, the broken bases of the basipterygoid processes project anteroventrally at approximately 45° (Fig. 1), as in flagellicaudatans and some titanosaurs[2,8,11]. Unlike most sauropods, but similar to dicraeosaurids[2,8], there is a deep slot-like channel on the ventral midline of the basisphenoid of *Lingwulong*, extending between the bases of the basipterygoid processes. IVPP V23704 preserves 29 teeth in a 'U'-shaped arc (Fig. 1, Supplementary Fig. 3), suggesting that the whole set either became detached from the jaw as a unit, or that jaw elements decayed, leaving the teeth in situ. This is probably a reliable indication that the jaw margin was not square, unlike those of most other diplodocoids[2,8,12]. We tentatively identify these as dentary teeth, based on distolabially facing wear facets: this would mean that there were ~15 dentary teeth in life. Wear facets resemble those of diplodocoids, being typically set at ≥45° to the apicobasal axis[7]. As in other diplodocoids, tooth crowns do not overlap each other in an imbricate arrangement[1,2]. The lingual surface of the crown is

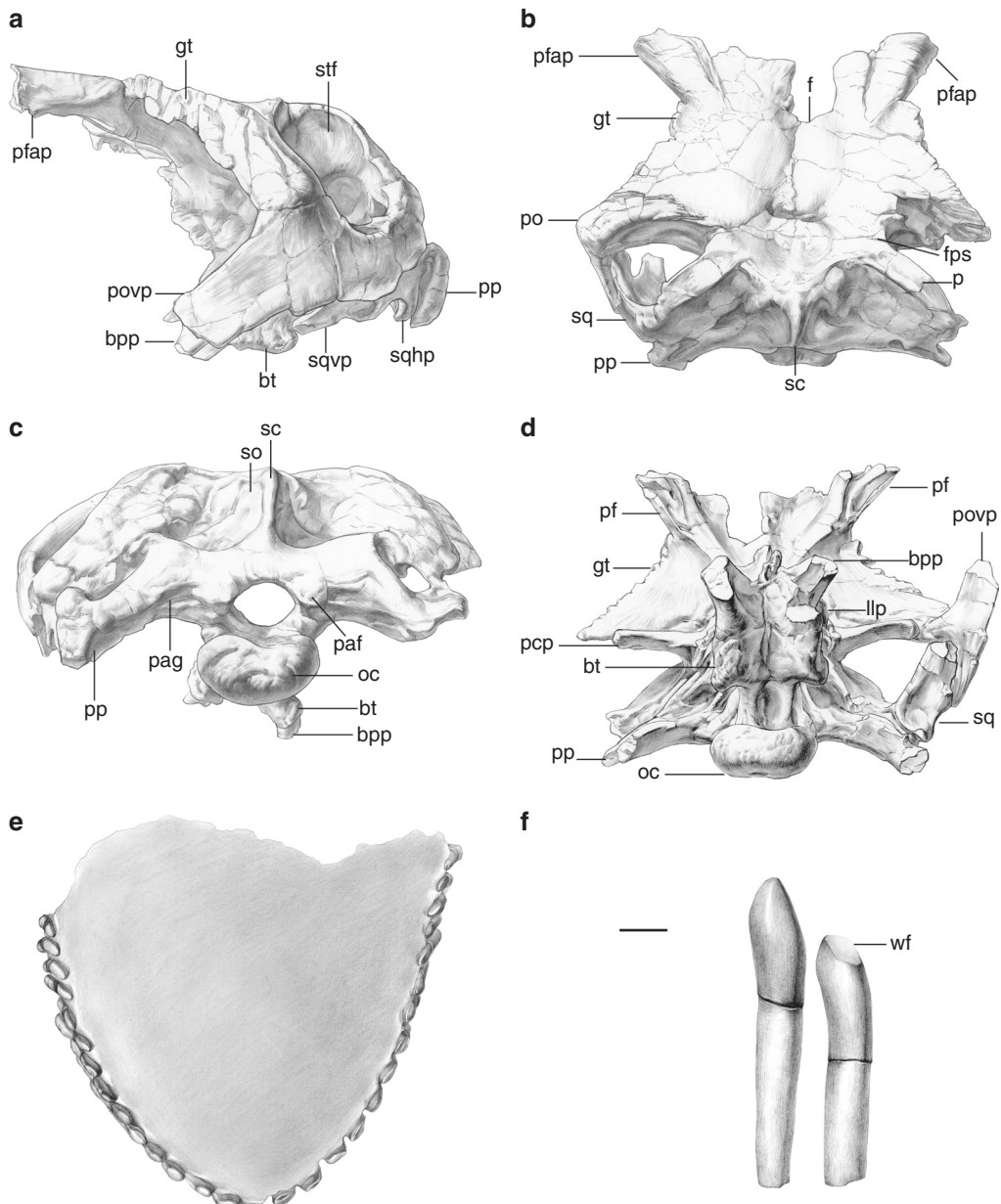

**Fig. 1** Cranial material of *Lingwulong shenqi*. Braincase in: left lateral (**a**), dorsal (**b**), occipital (**c**), and ventral (**d**) views. Dentary teeth in occlusal view (**e**). The 5th and 6th left dentary tooth crowns in labial view (**f**). Abbreviations: bpp, basipterygoid process; bt, basal tubera; f, frontal; fps, frontoparietal suture; gt, grooves and tubercles; pf, prefrontal; llp, 'leaf'-like process; oc, occipital condyle; p, parietal; pcp, capitate process; paf, proatlantal facet; pag, proatlantal groove; pfap, prefrontal anterior process; po, postorbital; povp, postorbital ventral process; pp, paroccipital process; sc, sagittal crest; so, supraoccipital; sq, squamosal; sqhp, squamosal hook-like process; sqvp, squamosal ventral process; stf, supratemporal fenestra; wf, wear facet. Scale bars = 20 mm for **a–e** and 10 mm for **f**

convex mesiodistally, creating an elliptical horizontal cross-section, a derived state that occurs convergently in diplodocoids and titanosaurs[1,2].

The true number of cervical, dorsal, and caudal vertebrae in *Lingwulong* is unknown, although we estimate 11–12 dorsals. As in dicraeosaurids[7], the ventral surfaces of the cervical centra are deeply excavated anteriorly to produce a pair of pneumatic fossae separated by a prominent midline keel (Supplementary Fig. 4). A deep lateral pneumatic opening is present on the anterior cervical centra, but is shallow or absent in more posterior cervical and dorsal vertebrae. Although the absence of deep lateral pneumatic openings in presacral centra is plesiomorphic for sauropods[1,2],

this also occurs as a derived reversal in dicraeosaurids[7]. In *Lingwulong* cervical centra, a small accessory fossa is located posteroventral to the main lateral pneumatic fossa (Supplementary Fig. 4). This feature has previously been reported only in diplodocids[7–9], but it is also variably present in the dicraeosaurid *Amargasaurus* (MACN N-15). Neural spines are bifurcated from the middle cervical vertebrae to approximately dorsal vertebra 5. Cervical neural spines lack the extreme elongation seen in more advanced dicraeosaurids such as *Dicraeosaurus* and *Amargasaurus*[7,13,14]. In lateral view (Fig. 2), there is a deep 'U'-shaped notch between the prezygapophyses and anterior spine margin, and the angle between the postzygodiapophyseal lamina

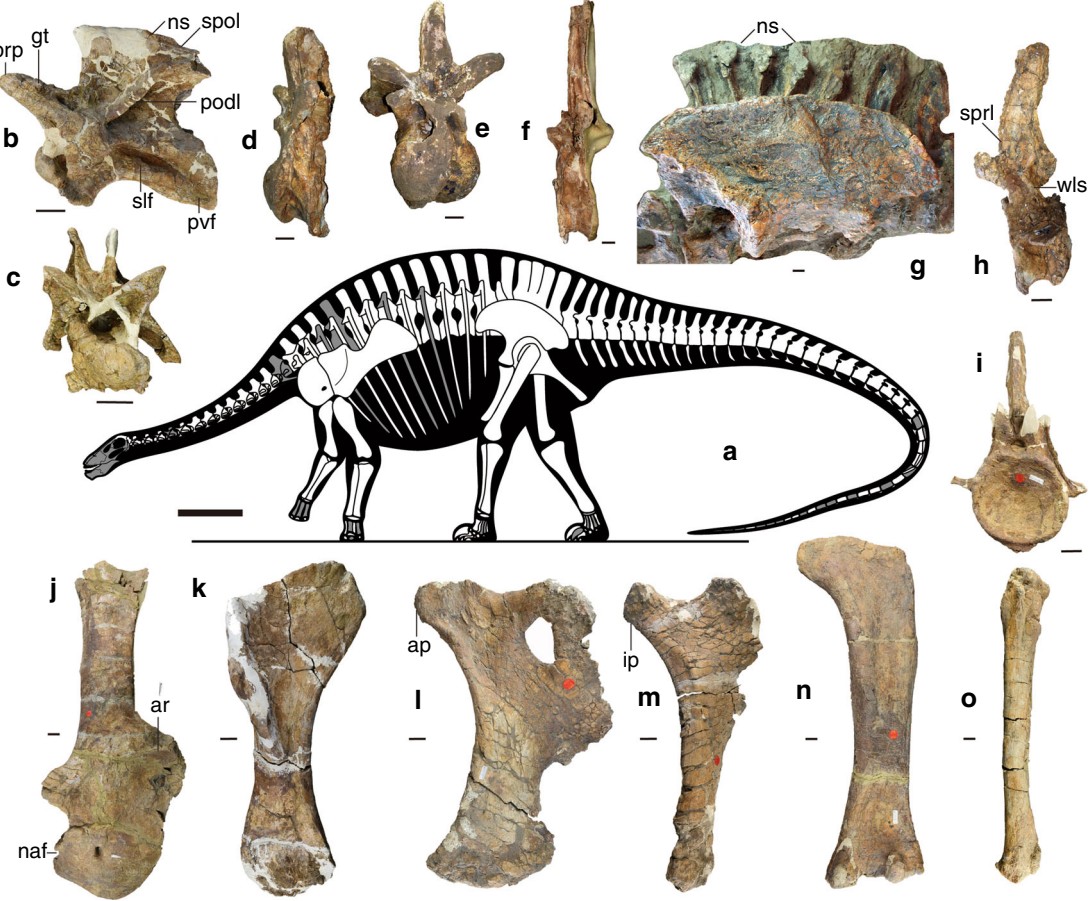

**Fig. 2** Skeletal reconstruction and exemplar skeletal remains of *Lingwulong shenqi*. Silhouette showing preserved elements (**a**); middle cervical vertebra in left lateral (**b**) and anterior (**c**) views; anterior dorsal vertebra in left lateral (**d**) and anterior (**e**) views; posterior dorsal vertebra in lateral view (**f**); sacrum and ilium in left lateral view (**g**); anterior caudal vertebra in left lateral (**h**) and anterior (**i**) views; right scapulocoracoid in lateral view (**j**); right humerus in anterior view (**k**); left pubis in lateral view (**l**); right ischium in lateral (**m**) views; right femur in posterior view (**n**); and right tibia in lateral view (**o**). Abbreviations: ap, ambiens process; ar, acromial ridge; ip, iliac peduncle; naf, notch anterior to glenoid; np, neural spine; podl, postzygodiapophyseal lamina; ppr, prezygapophyseal process ridge; prp, prezygapophysis; pvf, posteroventral fossa; slf, shallow lateral fossa; spol, spinopostzygapophyseal lamina; sprl, spinoprezygapophyseal lamina; wls, wing-like structure. Scale bars = 100 cm for **a** and 5 cm for **b**–**o**

(PODL) and spinopostzygapophyseal lamina (SPOL) is ~90°; both features being characteristic of dicraeosaurids[13,14]. The metapophyses of *Lingwulong* are directed dorsally to create a deep and transversely narrow 'V'-shaped notch, an intermediate condition between that seen in most sauropods with bifid presacral spines and the extremely tall and narrow notch in advanced dicraeosaurids[2,14]. *Lingwulong* possesses the derived shortened cervical ribs seen in other diplodocoids[7–9].

Anterior dorsal centra are strongly opisthocoelous, but from approximately dorsal vertebra 4 onwards they become amphicoelous, as in other diplodocoids[8]. Posterior dorsal neural spines are tall (spine:centrum height ~2.0), with spine height increasing towards the sacrum (Fig. 2). Such tall dorsal neural spines are a synapomorphy of Diplodocoidea, although in advanced dicraeosaurids (e.g., *Dicraeosaurus*) the spine:centrum height ratio increases to ~4.0[1,2,7–9]. In middle dorsal vertebrae, the spines have a 'paddle'-shaped morphology, in which their lateral margins gradually flare outwards as they approach the transversely rounded summit (Fig. 2, Supplementary Fig. 5), a derived condition seen in the middle and posterior dorsal spines of rebbachisaurids and dicraeosaurids[7,8,13]. In *Lingwulong*, the posterior-most dorsal spines more closely resemble those of non-diplodocoids (e.g., turiasaurs and non-titanosaurian

macronarians) in possessing well-developed subtriangular aliform processes that project laterally from the summit.

The sacrum (Fig. 2, Supplementary Fig. 6) comprises five fused vertebrae. Sacral centra lack lateral pneumatic fossae and are mildly amphicoelous. Sacral centra 3 and 4 are the most constricted transversely. Neural spines 2–4 are coalesced, as in most flagellicaudatans[2]. Sacral ribs 2–5 fuse distally to form a 'sacricostal yoke'[1], but only sacral ribs 2–4 actually contribute to the dorsal margin of the acetabulum.

Caudal centra are shallowly amphicoelous and subcircular in transverse cross-section. They lack lateral pneumatic fossae below the base of the rib, unlike some rebbachisaurids, diplodocines, and many titanosauriforms[7–9,15]. In anterior caudal vertebrae (Fig. 2, Supplementary Fig. 7), the low rounded spinoprezygapophyseal laminae (SPRLs) extend onto the lateral surface of each spine, a derived state observed in flagellicaudatans[1]. The first 11 caudal neural spines are unusual in having subtriangular facet-like areas on their lateral surfaces that extend from the summit to approximately spine mid-height. The ventral tips of these facets are expanded laterally to form small processes (Supplementary Fig. 7), resembling the triangular projections seen in some rebbachisaurids[8]. The anterior-most caudal ribs of *Lingwulong* have the wing-like structure present in most diplodocoids[2]: the

anterior surface is deeply excavated and the dorsolateral corner forms a distinct projection (Fig. 2, Supplementary Fig. 7). In most diplodocoids, however, the latter projection is a low 'shoulder'-like region, whereas in *Lingwulong* it is a prominent dorsally-directed prong. Mid-tail chevrons are 'forked', as occurs in non-neosauropod eusauropods and flagellicaudatans[1,16].

The scapula (Fig. 2) has a strongly expanded proximal plate and a long blade with a widened distal end (but it is not racket-shaped, in contrast to those of rebbachisaurids[8]). The prominent acromial ridge is at ~90° to the long-axis of the blade. In cross-section, the blade is 'D'-shaped, as is typical for all sauropods apart from early forms (e.g., *Shunosaurus*) and some advanced titanosaurs[1]. In the coracoid (Fig. 2), the notch anterior to the glenoid is weakly developed and the glenoid region does not expand markedly laterally, unlike those of some *Camarasaurus*-like macronarians[17]. The proximal end of the humerus (Supplementary Fig. 8) is strongly convex transversely (Fig. 2). As in diplodocids[9], the humerus is twisted so that, with the long-axis through the proximal end extending transversely, the long-axis through the distal end is directed posterolaterally. Ulnae possess a triradiate proximal end with a deep radial fossa (Supplementary Fig. 8), as in other sauropods[1,2]. The radius has a 'D'-shaped proximal end with a flattened rugose articular surface. The distal end of the radius is convex and, in anterior view, moderately beveled so that it faces laterodistally (Supplementary Fig. 8).

The ilium (Fig. 2) has a reduced ischial articulation, rounded dorsal profile, and lacks the brevis fossa, as is typical in all sauropods[2]. As in several other flagellicaudatans (e.g., *Dicraeosaurus*, *Diplodocus*[8]), the pubis bears a prominent ambiens process immediately anterior to the iliac articulation (Fig. 2): however, this process tapers in dorsoventral width towards its tip in *Lingwulong*, rather than being 'hooked'. The ischium is relatively slender. In lateral view (Fig. 2), the iliac peduncle of the ischium lacks the derived constriction ('neck') seen in most rebbachisaurids[7,8]. The distal end surface of the ischium retains the plesiomorphic subtriangular profile seen in most non-macronarian sauropods[2], yet the conjoined distal ends are coplanar (an apomorphy observed in macronarians and rebbachisaurids[1,2] that has not previously been seen in a flagellicaudatan).

The femur (Fig. 2, Supplementary Fig. 9) resembles those of other sauropods, although the proximal head is directed medially, rather than dorsomedially[2]. Its fourth trochanter is a low rounded ridge on the posteromedial margin of the femoral shaft, at approximately mid-length. The tibia (Fig. 2, Supplementary Fig. 9) has a transversely widened proximal end, as occurs in most neosauropods[1], with the cnemial crest directed laterally. Its distal end has a reduced medial malleolus, exposing the astragalus posteriorly, as in most sauropods[1]. The astragalus has a transversely and anteroposteriorly convex ventral surface and tapered medial projection. Its ascending process extends to the posterior margin of the astragalus, a derived state characteristic of neosauropods[1]. The fibular facet faces laterally, rather than posterolaterally as occurs in many diplodocoids[7].

Our phylogenetic analysis places *Lingwulong* within Dicraeosauridae as the sister taxon to *Amargasaurus* + (*Brachytrachelopan* + *Dicraeosaurus*) (Supplementary Note 4, Fig. 3, Supplementary Figs. 10–12, 15, Supplementary Data 15). Symmetric resampling indicates that this placement of *Lingwulong* is the sixth best-supported node out of the 70 nodes in the tree (Supplementary Fig. 11).

In our biogeographic analyses (Supplementary Note 6), the log likelihood ratio tests demonstrate that the +J versions of the biogeographic models are very strongly significantly better fits to the data than are the non+J versions (*p*-values range from 5.3e$^{-8}$ to 4.4e$^{-26}$: see Supplementary Data 16). Moreover, the AIC

values for BAYAREALIKE + J are 18.7 (relaxed) and 23 (harsh) units lower than the next best supported model (i.e., DEC+J), which suggests that the former model can be regarded as strongly outperforming the other five models[18]. The results of our additional analyses of a reduced dataset were very similar to those produced by the analyses of the full dataset, with BAYAREALIKE+J being strongly preferred over other models (Supplementary Data 17).

## Discussion

*Lingwulong* possesses 22–23 synapomorphies (10–11 of which are unequivocal) that support placement within Diplodocoidea or a less inclusive clade (Supplementary Data 15). For example, synapomorphies uniting *Lingwulong* with other dicraeosaurids include: coalesced frontals; a deep slot-like region between the bases of the basipterygoid processes; a right-angle between the cervical PODL and SPOL; and loss of pneumatic openings in dorsal centra. *Lingwulong* also possesses several unequivocal synapomorphies of Flagellicaudata (e.g., a prominent ambiens process on the pubis) and Diplodocoidea (e.g., the SPRLs of anterior caudal vertebrae extending onto the lateral surface of the neural spine) (Supplementary Data 15). The lack of some dicraeosaurid features (e.g., the greatly elongated cervical metapophyses seen in *Amargasaurus*) is consistent with the 'basal' position of *Lingwulong* within this clade and its early stratigraphic age. *Lingwulong* also displays a few plesiomorphies that are inconsistent with diplodocoid affinities, including: frontal contributing to the supratemporal fenestra; 'U'-shaped mandible in dorsal view; few replacement teeth per alveolus; and a laterally facing fibular facet on the astragalus. However, these apparent reversals are relatively rare, and might indicate that some aspects of diplodocoid feeding strategies or locomotion evolved independently several times within this clade. Moreover, a more 'U'-shaped snout is consistent with the placement of *Lingwulong* as a 'basal' dicraeosaurid (i.e., outside of the *Dicraeosaurus-Amargasaurus-Brachytrachelopan* clade), given that this family tends to have less 'square' snouts than rebbachisaurids and diplodocids[12].

Despite occasional doubts[19], the East Asian Isolation Hypothesis (EAIH) has become a well-established explanation of profound differences between Jurassic (and sometimes Early Cretaceous) Asian terrestrial faunas and those elsewhere in Pangaea[6]. The EAIH postulates that isolation of East Asia[3–6,20–26] resulted in the evolution of endemic groups such as mamenchisaurid sauropods[27], an early diverging lineage of tetanurans[28], oviraptorosaurs[4], therizinosaurs[4], the initial neornithischian radiation[29], marginocephalians[4], and some mammals[4,30]. The favored isolating mechanism is an epicontinental seaway located west of the Urals, separating Central and East Asia from the rest of Laurasia[3–6,20–25,31] (Supplementary Note 6; Fig. 4). This barrier has also been used to explain the absence in East Asia of many groups that were present elsewhere in Pangaea during the Jurassic, including diplodocoid and titanosauriform sauropods, dromaeosaurs, nodosaurids, and the lineage leading to iguanodontian ornithopods[3–6,20,26,30,31]. According to this scenario, isolation ended in the Early Cretaceous when marine regressions allowed the invasion of groups from elsewhere in Pangaea (e.g., titanosauriforms, dromaeosaurs, ornithomimosaurs, iguanodontians, gekkonid, and paramacellodid lizards [though see ref.[32]], and several mammalian families), and also the dispersal of Asian endemics (e.g., oviraptorosaurs, marginocephalians) into Europe and/or North America[3–6,20,23,27,33–35]. It has been claimed, however, that diplodocoids never took part in these dispersals[5,6] because of an end-Jurassic extinction that greatly reduced their diversity and geographic range in the Early Cretaceous[7,8,36].

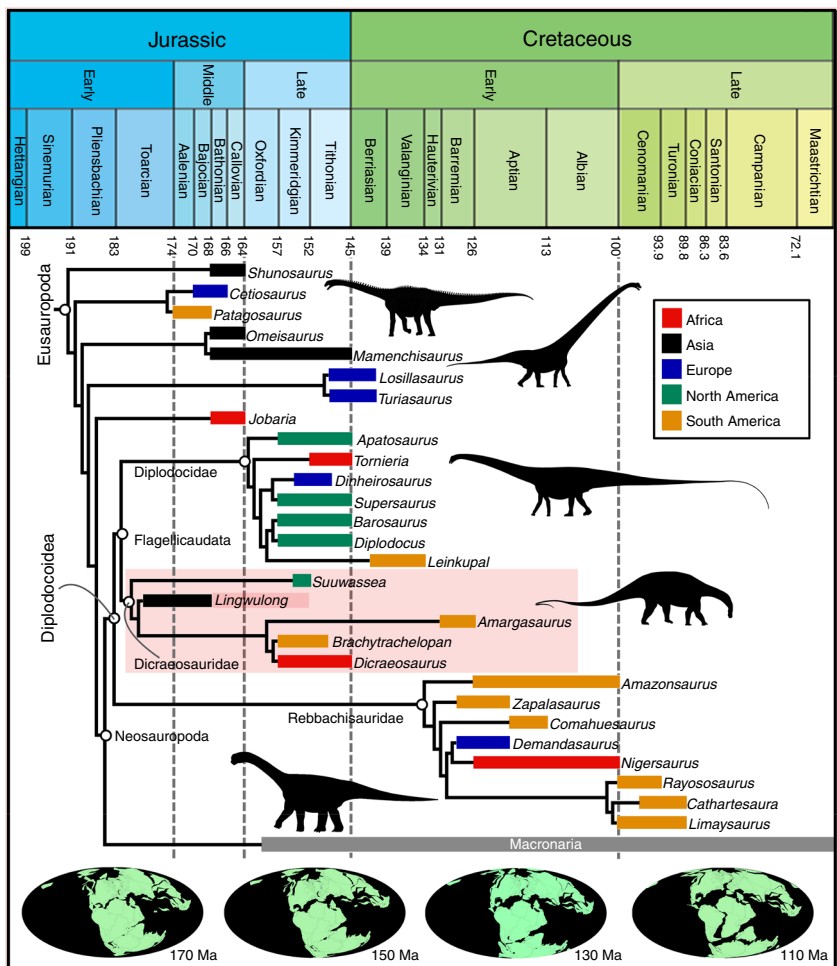

**Fig. 3** Time-calibrated evolutionary tree for Eusauropoda. Agreement subtree produced in TNT, with additional diplodocid taxa incorporated (see Supplementary Note 4). All macronarian taxa have been combined into a single lineage, and non-sauropod sauropodomorphs have been removed, in order to enhance clarity (see Supplementary Fig. 13 for the full version of this tree). Silhouettes of dinosaurs drawn by Scott Hartman, Mike Taylor, and Mathew Wedel, and available at Phylopic (http://phylopic.org/) under a Creative Commons Attribution 3.0 Unported license (https://creativecommons.org/licenses/by/3.0/). Global paleogeographic reconstructions from the Paleobiology Database (https://www.paleobiodb.org)

The discovery of an early Middle Jurassic Chinese dicraeosaurid, and the results of our biogeographic analyses using the Maximum Likelihood R package BioGeoBEARS (Supplementary Notes 2, 5, and 6), contradict several aspects of the EAIH. At least one diplodocoid lineage had reached East Asia by the Middle Jurassic, and ancestral range estimations (Supplementary Note 6) suggest that major clades such as Neosauropoda, Macronaria, Diplodocoidea, Flagellicaudata, and Diplodocidae, were widespread across Pangaea (including East Asia) by the early Middle Jurassic. This is consistent with previous suggestions that, despite a lack of direct fossil evidence, many dinosaur groups with global distributions in the Late Jurassic and/or Cretaceous, probably originated before the separation of Laurasia from Gondwana during the Bathonian–Callovian[3,37,38].

Recently, a diluted version of the EAIH has been proposed[27], in which macronarians and non-neosauropod eusauropods were present in China during the Middle Jurassic, but regional extinctions and the onset of geographic isolation caused Late Jurassic East Asian faunas to lose their neosauropods and become dominated by endemic mamenchisaurids. Although there is paleogeographic evidence for a limited period of East Asian isolation during the late Callovian–early Kimmeridgian

(Supplementary Note 7; Fig. 4), problems remain with this revised EAIH. Refined dating of key sedimentary units in China pushes back many of the mamenchisaurid-dominated faunas into the Middle Jurassic[27] (i.e., before geographic isolation occurs), and places the putative macronarian *Bellusaurus* at the Middle/Late Jurassic boundary, potentially in the early Late Jurassic (Supplementary Note 7). Moreover, mamenchisaurids might not have been endemic to East Asia, given that a specimen from the Late Jurassic of Tanzania was referred to this clade recently[11]. Thus, we suggest that: (1) both macronarians and diplodocoids were present in East Asia continuously from the early Middle Jurassic onwards; and (2) the absence of these neosauropods in the Late Jurassic of East Asia is more plausibly interpreted as reflecting heterogeneous spatial and temporal sampling (Supplementary Note 7). The sampling-based explanation for the absence of neosauropods in the Jurassic of East Asia has previously been rejected on the grounds that this region has extensive sauropod-rich Jurassic sediments, all dominated by non-neosauropods[5,6,26,33]. However, sampling is worse in the Late Jurassic than the Middle Jurassic[27], and the mamenchisaurid-dominated deposits are clustered in southwest China rather than the northwestern region that yielded *Lingwulong* (Supplementary Note 7). In short, the discovery of *Lingwulong* undermines the

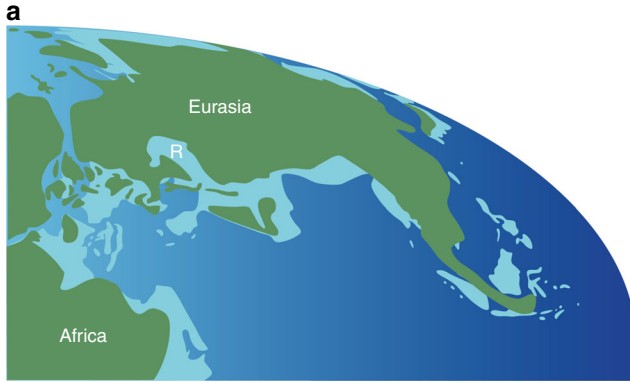

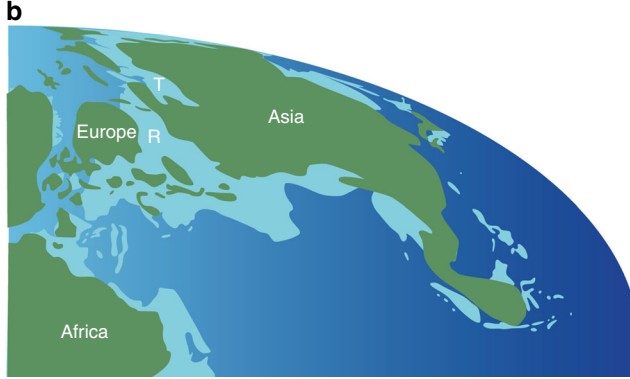

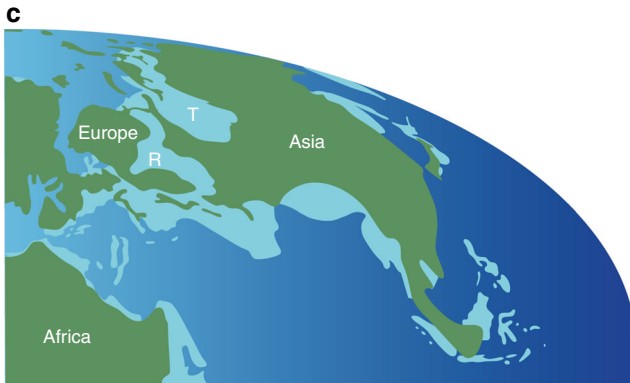

**Fig. 4** Paleogeographic maps showing the formation and disappearance of an epicontinental seaway between Europe and Central Asia during the Middle Jurassic through Early Cretaceous. **a** Middle Jurassic (170 Ma); **b** Late Jurassic (160 Ma); **c** Early Cretaceous (138 Ma). The maps are based on ref. [48]. Green indicates land, light blue shallow sea, and deep blue ocean. Abbreviations: R, Russian Platform Sea; T, Turgai Sea

EAIH and suggests that critical re-assessments of the established biogeographic histories of other implicated groups (e.g., theropods, ornithischians, pterosaurs, lizards, and mammals) are warranted.

The estimated 174 Ma age for *Lingwulong* (Supplementary Notes 2 and 5) means that it is the earliest diplodocoid and therefore also the earliest neosauropod. This discovery necessitates a significant revision of hypotheses concerning the origins and early radiation of Neosauropoda. Previously, the earliest accepted neosauropods came from the early Late Jurassic (Supplementary Note 5). The fossil record has implied that, although neosauropods probably originated in the Middle Jurassic, they did not become widespread and diverse until the Late Jurassic. For example, it has been proposed that neosauropods radiated and replaced many non-neosauropod eusauropods during a 5 million year window close to the Middle/Late Jurassic boundary[26].

However, our phylogenetic and biogeographic results make the hypothesized rapid diversification at the Middle/Late Jurassic boundary difficult to maintain. Neosauropods were probably diverse and widespread as early as the early Middle Jurassic (Fig. 3, Supplementary Fig. 13), so the perceived increase in their dominance around the Middle/Late Jurassic boundary might reflect an increase in relative abundance and/or a shift to habitats with higher preservation potentials, rather than a genuinely rapid diversification at this time.

Placement of *Lingwulong* within the diplodocoid clade Dicraeosauridae results in predicted range extensions of > 15 Myr for Rebbachisauridae, Diplodocidae, the lineages leading to *Suuwassea* and 'advanced' dicraeosaurids (i.e., *Dicraeosaurus*, *Amargasaurus*, etc.), Macronaria, and the non-neosauropod turiasaurs (Fig. 3, Supplementary Fig. 13). This suggests that the Middle Jurassic diversity of neosauropods was substantially higher than previously estimated, but that this diversity is currently cryptic due to considerable sampling biases. Although anthropogenic and geologic biases and their effects on the dinosaur fossil record have been noted before (e.g. ref.[36]), this new discovery shows that these biases might have been even more severe, on a global scale, than previously thought. Moreover, since the oldest known dicraeosaurid occurs in the early Middle Jurassic, then major sauropod clades such as Neosauropoda, Macronaria, Diplodocoidea, Rebbachisauridae, Flagellicaudata, Dicraeosauridae, and Diplodocidae are likely to have originated during the Early Jurassic, far earlier than previously realized[7,26,36]. This 'pushing back' of the origination times of major sauropod clades reinforces recent suggestions that the Early Jurassic was a critical phase in dinosaur evolution, characterized by highly elevated rates of diversification and morphological change[38–40].

## Methods

**Phylogenetic analysis**. Phylogenetic analyses were carried out in TNT vs. 1.5[41]. The New Technology Search was applied first, using sectorial searches, ratchet, drift, and tree fusing, with the consensus stabilized 10 times. The resulting MPTs were then used as the starting trees for a Traditional Search using TBR. The support for each node in the trees was assessed in TNT using GC values generated via symmetric resampling, based on 5000 replicates[42]. The latter analyses used the Traditional Search option with TBR. Character mapping was carried out in Mesquite version 2.75[43]. In order to examine the possibility that *Lingwulong* is not a neosauropod, the 'force' command was used in TNT to impose such a constraint. The resulting trees were then compared with the original (i.e., unconstrained) MPTs using a Templeton's test implemented in PAUP 4.0[44].

**Biogeographic analysis**. The biogeographic analyses using the R package BioGeoBEARS[45,46] require a dated phylogenetic topology (Supplementary Note 6). BioGeoBEARS enables ancestral area estimation using Maximum Likelihood[45,46]. It implements six different models: DEC, DEC+J, DIVALIKE, DIVALIKE+J, BAYAREALIKE, and BAYAREALIKE+J. DEC and DIVALIKE allow different forms of vicariance to occur at nodes, whereas BAYAREALIKE disallows vicariance and instead forces daughter lineages to inherit the range of their immediate ancestor[45]. The +J versions of each model have the same properties as their non+J versions, except that the former also allow founder-event speciation to occur at ancestral nodes[45,46]. Log likelihood ratio tests and AIC values are used to determine which of these models best fits the data. In our analyses, we allowed each ancestor to occupy up to the full eight geographic areas available. We ran analyses, using the relaxed and harsh versions of our dispersal multiplier matrices. BioGeoBEARS was run in R version 3.2.3[47], and the script used is presented in Supplementary Data 9. One potential drawback with our biogeographic dataset is that we cannot be sure that the relationships of the four additional diplodocids grafted onto our agreement subtree would be supported if they had been incorporated into our phylogenetic analysis. Therefore, in order to test the impact of these four taxa on our biogeographic conclusions, we re-ran our BioGeoBEARS analyses using the same protocol outlined above, but with these taxa removed.

**Nomenclatural acts**. This published work and the nomenclatural acts it contains have been registered in ZooBank, the proposed online registration system for the International Code of Zoological Nomenclature (IZCN). The ZooBank LSIDs (Life science identifiers) can be resolved and the associated information viewed through any standard web browser by appending the LSID to the prefix "http://zoobank.

org/". The LSID for this publication is urn:lsid:zoobank.org:pub:F28E9AD2-2784-4923-AA59-3A6D3CDF51A0.

**Data availability**. The data reported in this publication are detailed in the main text and its supplementary files.

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

## Acknowledgements

We thank Haijun Wang, Lishi Xiang, He Sicai, Renfang Cao, Zhilu Tang, and Yu Tao for excavating and preparing specimens, and Aijuan Shi (Figs. 1e, 1f, 2a, and 4, Supplementary Fig. 1), Hailong Zang (all photographs), and Rongshan Li (Fig. 1a–d) for illustrations. This work was supported by grants from the National Natural Science Foundation of China (91514302, 41688103, and 41120124002), the Strategic Priority Research Program of the Chinese Academy of Sciences (Grant No. XDB18030504), Lingwu City Government, National Geographic Society's Waitt Grant Scheme (award no. W421-16), the Earth Sciences DIF of the Natural History Museum, a Leverhulme Trust Early Career Fellowship (ECF-2014-662), and a Royal Society University Research Fellowship (UF160216). We are grateful to the Willi Hennig Society for their support of the phylogenetics package TNT.

## Author contributions

X.X., P.U., and P.D.M. designed the project. All authors performed the research. X.X., P.U., P.D.M., and P.M.B. wrote the manuscript.

## Additional information

**Competing interests:** The authors declare no competing interests.

