## [Peer Review File · Nature Communications]

Reviewers' comments:

Reviewer #1 (Remarks to the Author):

The manuscript describes a new dicraeosaurid sauropod from Asia. The new species is not solely relevant due to the low account of species known from this clade, and therefore helping into better understand the diversity of the clade, but also due to the great impact that this new species has on both, the evolution of Neosauropoda (the main clade of Sauropoda) and the biogeographic history of Pangea, specially Laurasia and the isolation of Asia. I just found the paper of extreme interest, and after a couple of lectures I only have one minor comment related to the abbreviation of figure 1 (the marked frontal seems to be the prefrontal).

The new species is clearly described, and a broader description is provided in the supp info. This is an important point, as the basic anatomical information is available for sauropod specialist. But the manuscript clearly has a broad focus, which is of major importance for other non-dinosaur specialist, and even for non-paleontologist, as a new biogeographic scenario, related to the isolation of Asia, is proposed. The analyses supporting both, the phylogenetic placement of the new dicraeosaurid and the biogeographic scenario are appropriated and well explained in the text and supplementary material.

I found the paper of the exact match that a journal like this is looking for, as will have a broad repercussion in the community. My personal recommendation is that the manuscript could be accepted as is.

Reviewer #2 (Remarks to the Author):

Overall, this is a well-written and thorough manuscript with certainly intriguing results on this new (neo)sauropod from the Jurassic of China. There is nothing overtly fatal or glaring in the study as far as I can tell to suggest anything beyond minor revisions. If the analyses and interpretations are supported in future studies (I do look forward to the long description, too), then this is indeed a significant discovery for our understanding of sauropods and dinosaur diversification and paleobiogeography in the Early-Mid Jurassic. It also helps there are multiple specimens and individuals with most of the skeleton to study and greatly inform their analyses and interpretations! The morphological description is supported by various synapomorphies and autapomorphies that substantiate their claims for a new and earliest neosauropod dinosaur with diplodocoid affinities. A surprising claim given the handful of previously proposed but unsubstantiated diplodocoids from the region which the authors address in the SI. The analyses themselves are straightforward but I would like to make a suggestion or two to help potentially improve their interpretations. Please check out my comments in the PDFs for further edits and suggestions.

1. Figures. Generally, the figures are decent and the figures for the main document could be vastly improved with labels. You don't have to label every bump and notch; however, I would strongly recommend labeling key diplodocoid/dicraeosaurid synapomorphies discussed in the description as well as all (if feasible) autapomorphies.

2. Stay consistent with basal/derived and related terms. Taxa are not basal/derived as they are composed of a mixture of derived and plesiomorphic characters. This inaccurate use of the terms has been persistent in the paleo literature so I understand what is being conveyed. Please check throughout the manuscript for these instances.

(See: Mario Bronzati. 2017. Should the terms 'basal taxon' and 'transitional taxon' be extinguished from cladistic studies with extinct organisms? *Palaeontologia Electronica* 20:1–12.)

3a. I would like to see the results of the BioGeoBEARS/paleobiogeographical analysis without the four 'grafted' taxa (e.g., Tornieria, Dinheirosaurus, Supersaurus, and Leinkupal) and see how that differs from their addition. Since they are not in your data set for the phylogenetic analyses, there is the slight chance they may not appear where they were in the Tschopp et al analyses which would then potentially alter the results. Either add them into the phylogenetic analysis (which, understandably, would take additional time and effort to score and run) or do what I suggested above (a more reasonable recommendation). I would also strongly recommend adding the BioGeoBEARS reconstruction results into the SI figures.

3b. Following up on the paleobiogeographical analyses, I find it interesting that BAYAREALIKE(+j) is the favorable model. I've noticed this is similar to the Poropat et al 2016 study but differs from the Gorscak and O'Connor 2016 study where they recovered the BAYAREALIKE(+j) models as the least-supported models. Considering these recent paleobiogeographic studies also pertain to sauropods (generally), it should be worth noting the differences among these studies. This isn't too surprising as the focus of the current study is of a time when the landmasses were coherent and a sympatric model may, in fact, be the likeliest explanation for the pattern whereas the DEC/DIVALIKE(+j) models (vicariance-friendly, if you will) were supported in Gorscak and O'Connor study that focused on a time when the landmasses were breaking apart from one another.

4. I think the authors should try to implement the new taxon into a recent diplodocoid-focused phylogenetic matrix too (this could be saved for the SI as I doubt it will drastically change their interpretations for it being a diplodocoid of sorts). The Rauhut matrix is great for a sauropod-wide sampling and fits the narrative for pushing neosauropod origins earlier. However, I am curious if these results will hold true in a more specific data set on diplodocoids... if Lingwulong will still hold as a dicraeosaur or potentially as some stem lineage around the base of diplodocoids.

5. There are many typos and inconsistent formatting in the references for both the main document and the SI. Please fix accordingly.

I look forward to the final publication of this important study and I wish the authors luck in their revisions!

Summary of Comments from Reviewer 2 on Article

Page: 1

Author: I am a Reviewer Date: Indeterminate

I would suggest starting broad first (eg, tetrapod groups) then getting more specific (diplodocids). Since this is the first time dipodocids are mentioned, I would suggest saying something to the effect of "diplodocid sauropod dinosaurs". 鼎

Author: redacted Date: Indeterminate

I would suggest starting broad first (eg, tetrapod groups) then getting more specific (diplodocids). Since this is the first time dipodocids are mentioned, I would suggest saying something to the effect of "diplodocid sauropod dinosaurs". 鼎

Author: I am a Reviewer Date: Indeterminate

Since diplodocid sauropod is already established, you could save some space by omitting sauropod here.

Author: redacted Date: Indeterminate

repeated from previous clause.

Author: redacted Date: Indeterminate

Author: I am a Reviewer Date: Indeterminate

Dispersal of which groups? Tetrapods, diplodocids, sauropods?

Author: redacted Date: Indeterminate

Dispersal of which groups? Tetrapods, diplodocids, sauropods?

Page: 2

 Author: redacted Date: Indeterminate
Neosauropodan

Page: 3

Author: redacted Date: Indeterminate

This goes with my comments on the anatomical figures. I would strongly recommend labeling the autapomorphies, and several key synapomorphies, in the figures.

Page: 4

Author: redacted Date: Indeterminate

Author: redacted Date: Indeterminate

 Author: redacted Date: Indeterminate

Author: redacted Date: Indeterminate

Here and elsewhere, you do not need "In X View". A process is anteromedially directed no matter the perspective. Only use "In X View" if you are describing a contour or shape that can only be seen in that view... even then, there are likely better ways of describing morphology instead of relying on perspective views.

 Author: redacted Date: Indeterminate

Page: 5

Author: redacted Date: Indeterminate

Isn't a reversal by definition a derived condition? This is redundant, just use reversal.

Author: redacted Date: Indeterminate

Author: redacted Date: Indeterminate

Author: redacted Date: Indeterminate

Author: redacted Date: Indeterminate

Page: 6

Author: redacted Date: Indeterminate

Author: redacted Date: Indeterminate

Author: redacted Date: Indeterminate

Author: redacted Date: Indeterminate

Author: redacted Date: Indeterminate

How is it constricted? Transversely? anteroposteriorly? Circumferentially?

Page: 7

Author: redacted Date: Indeterminate

Some of these synapomorphies and the others following would be helpful if labeled in the figures.

Author: redacted Date: Indeterminate

Author: redacted Date: Indeterminate

Just to clarify, for the entire Jurassic or part of the Jurassic? If it is the latter part of the Jurassic, then you would still be consistent with this hypothesis if the Lingwulong branch occurred before EAIH took into effect.

Page: 8

Author: redacted Date: Indeterminate

Author: redacted Date: Indeterminate

An expansion of the first interpretation would be that neosauropods were present but rare components in East Asian faunas.

Author: redacted Date: Indeterminate

Page: 9

Author: redacted Date: Indeterminate

Author: redacted Date: Indeterminate

Re-assessment of previously collected materials that take into account traditionally non-East Asian tetrapod groups or re-assessment based on new body fossils for these groups in Early/Mid Jurassic East asia deposits?

Author: redacted Date: Indeterminate

Author: redacted Date: Indeterminate

Just replace with Dicreosauridae

Author: redacted Date: Indeterminate

Please check the references, there are several errors throughout (italics, typos, etc).

Page: 10

Author: redacted Date: Indeterminate

Author: redacted Date: Indeterminate

Author: redacted Date: Indeterminate

Author: redacted Date: Indeterminate

Author: redacted Date: Indeterminate

Pages? Issue?

Author: redacted Date: Indeterminate

Author: redacted Date: Indeterminate

Page: 11

Author: redacted Date: Indeterminate

Author: redacted Date: Indeterminate

Author: redacted Date: Indeterminate

Author: redacted Date: Indeterminate

Author: redacted Date: Indeterminate

Page: 12

Author: redacted Date: Indeterminate

Author: redacted Date: Indeterminate

Author: redacted Date: Indeterminate

Page: 13

Author: redacted Date: Indeterminate

Is it me or is the font size different here? Please have consistency in fonts through out manuscript

Page: 14

Author: redacted Date: Indeterminate

Great drawings!

Author: redacted Date: Indeterminate

Great drawings, curious, but why use the drawings versus the photos of these in the supplemental material?

Author: redacted Date: Indeterminate

I only see one scale bar...

Author: redacted Date: Indeterminate

Add period

Author: redacted Date: Indeterminate

Labels please

Author: redacted Date: Indeterminate

Version?

Author: redacted Date: Indeterminate

Author: redacted Date: Indeterminate

Page: 18

 Author: redacted Date: Indeterminate

This white left margin should be removed much like the right side of the figure.

Summary of Comments from Reviewer 2 on Supplementary Information

Page: 7

Author: redacted Date: Indeterminate

Author: redacted Date: Indeterminate

Author: redacted Date: Indeterminate
Unusual compared to which taxa? and/or what is the usual condition?

Author: redacted Date: Indeterminate

Author: redacted Date: Indeterminate

Author: redacted Date: Indeterminate

Author: redacted Date: Indeterminate

Page: 8

Author: redacted Date: Indeterminate

Author: redacted Date: Indeterminate
symmetrical in which way? mesiodistally? labiaolingually? apicobasally? ;-)

 Author: redacted Date: Indeterminate

 Author: redacted Date: Indeterminate

 Author: redacted Date: Indeterminate

Author: redacted Date: Indeterminate

Author: redacted Date: Indeterminate

Page: 9

Author: redacted Date: Indeterminate

Page: 10

Author: redacted Date: Indeterminate

Author: redacted Date: Indeterminate

Author: redacted Date: Indeterminate

Author: redacted Date: Indeterminate

Author: redacted Date: Indeterminate

Page: 11

Author: redacted Date: Indeterminate

Author: redacted Date: Indeterminate

Author: redacted Date: Indeterminate

Page: 12

Author: redacted Date: Indeterminate

Most posterior caudal vertebrae? most posterior dorsal vertebrae? Please specify

Page: 14

Author: redacted Date: Indeterminate

Page: 15

Author: redacted Date: Indeterminate

Oh thank goodness! Otherwise this could have been a mess to deal with.

Page: 16

Author: redacted Date: Indeterminate

 Author: redacted Date: Indeterminate

Author: redacted Date: Indeterminate

Page: 17

Author: redacted Date: Indeterminate

Author: redacted Date: Indeterminate

By that logic, wouldn't any one taxon within Diplodocoidea be susceptible?

Page: 18

Author: redacted Date: Indeterminate

Page: 19

Author: redacted Date: Indeterminate

Page: 20

Author: redacted Date: Indeterminate

Author: redacted Date: Indeterminate

single quotations not necessary

Page: 21

Author: redacted Date: Indeterminate

Page: 23

Author: redacted Date: Indeterminate

How exactly did you treat it as such? What dispersal multiplier was assigned here?

Author: redacted Date: Indeterminate

Perhaps moving this section of the SI earlier in 6.4. Dispersal multiplier matrices will help with what you mean by "harsh" and "relaxed" that have been mentioned in the preceding paragraphs. Get the definitions out of the way first, then go into your dispersal settings/justifications afterwards.

Page: 24

Author: redacted Date: Indeterminate

add a space between

Page: 25

Author: redacted Date: Indeterminate

Page: 26

Author: redacted Date: Indeterminate

Author: redacted Date: Indeterminate

Author: redacted Date: Indeterminate

is this suppose to say "low abundance OF habitat preference"? the and/or and or makes this sentence a touch hard to understand, re-word to clarify.

Page: 37

Author: redacted Date: Indeterminate

For consistency across figures, remove 8cm by the scale bar in the figure here.

Page: 40

Author: redacted Date: Indeterminate

Drop the grey background in C

Author: redacted Date: Indeterminate

remove underline

Page: 41

Author: redacted Date: Indeterminate

Remove the very lightly present gray background in a-c. You may need to adjust brightness/contrast to see it, I can definitely see it.

Page: 43

Author: redacted Date: Indeterminate

OTU, right?

Page: 47

Author: redacted Date: Indeterminate

As in the main document, there are many errors and typos in the references. I have highlighted what I noticed, but please carefully check each reference so that it is formatted properly.

Author: redacted Date: Indeterminate

Author: redacted Date: Indeterminate

Author: redacted Date: Indeterminate

Author: redacted Date: Indeterminate

Author: redacted Date: Indeterminate

Author: redacted Date: Indeterminate

Page: 48

Author: redacted Date: Indeterminate
Check font

Author: redacted Date: Indeterminate

Author: redacted Date: Indeterminate

Author: redacted Date: Indeterminate

Author: redacted Date: Indeterminate

Author: redacted Date: Indeterminate

Author: redacted Date: Indeterminate

Author: redacted Date: Indeterminate

Author: redacted Date: Indeterminate

Author: redacted Date: Indeterminate

Author: redacted Date: Indeterminate

Author: redacted Date: Indeterminate

Page: 49

Author: redacted Date: Indeterminate

Author: redacted Date: Indeterminate

Author: redacted Date: Indeterminate

Author: redacted Date: Indeterminate

Author: redacted Date: Indeterminate

Author: redacted Date: Indeterminate

Author: redacted Date: Indeterminate

Author: redacted Date: Indeterminate

Author: redacted Date: Indeterminate

Page: 50

Author: redacted Date: Indeterminate

Author: redacted Date: Indeterminate

Author: redacted Date: Indeterminate

Author: redacted Date: Indeterminate

Author: redacted Date: Indeterminate

Author: redacted Date: Indeterminate

Author: redacted Date: Indeterminate

Author: redacted Date: Indeterminate

Page: 51

Author: redacted Date: Indeterminate

Author: redacted Date: Indeterminate

Reviewer 1

"...I just found the paper of extreme interest, and after a couple of lectures I only have one minor comment related to the abbreviation of figure 1 (the marked frontal seems to be the prefrontal)."

Done. We have moved the label.

Reviewer 2

1. *"Figures. Generally, the figures are decent and the figures for the main document could be vastly improved with labels. You don't have to label every bump and notch; however, I would strongly recommend labeling key diplodocoid/dicraeosaurid synapomorphies discussed in the description as well as all (if feasible) autapomorphies."*

Done. We have added more labels to Figs 1, 2, and S4–9.

2. *"Stay consistent with basal/derived and related terms. Taxa are not basal/derived as they are composed of a mixture of derived and plesiomorphic characters. This inaccurate use of the terms has been persistent in the paleo literature so I understand what is being conveyed. Please check throughout the manuscript for these instances. (See: Mario Bronzati. 2017. Should the terms 'basal taxon' and 'transitional taxon' be extinguished from cladistic studies with extinct organisms? Palaeontologia Electronica 20:1 - 12.)"*

Done. We have replaced the term 'basal' throughout our main text and supplementary files. This required us to use slightly more cumbersome expressions, e.g., instead of 'basal eusauropod' we say 'non-neosauropod eusauropod'. Removing the term 'derived' with regard to taxonomic groups is more difficult. However, we have dealt with this issue through a combination of replacing 'derived' with 'advanced', and by specifying the contents of particular groups. Thus, for example, instead of 'derived dicraeosaurids', we simply specify the three genera which comprise this grouping.

3.a. *"I would like to see the results of the BioGeoBEARS/paleobiogeographical analysis without the four 'grafted' taxa (e.g., Tornieria, Dinheirosaurus, Supersaurus, and Leinkupal) and see how that differs from their addition. Since they are not in your data set for the phylogenetic analyses, there is the slight chance they may not appear where they were in the Tschopp et al analyses which would then potentially alter the results. Either add them into the phylogenetic analysis (which, understandably, would take additional time and effort to score and run) or do what I suggested above (a more reasonable recommendation). I would also strongly recommend adding the BioGeoBEARS reconstruction results into the SI figures."*

Done. We have re-run our biogeographic analyses with the four grafted taxa omitted. The results have been added to the Supplementary Material. Although there are some minor changes to the ancestral area estimations for some of the nodes, all of the key results remain essentially unchanged and therefore have no impact on our original conclusions.

For pragmatic reasons, we have not added figures for our biogeographic outputs to the Supplementary Materials. The outputs from the analysis are large PDF-based diagrams that extend over several pages. These would be difficult for readers to examine if they were incorporated into the Supplementary Materials. Therefore, we provide these biogeographic outputs as separate auxiliary Supplementary Material files that are referenced in the text. Thus, the reader will be made aware of them and able to access them more easily.

3.b.. *"Following up on the paleobiogeographical analyses, I find it interesting that BAYAREALIKE(+j) is the favorable model. I've noticed this is similar to the Poropat et al 2016 study but differs from the Gorscak and O'Connor 2016 study where they recovered the BAYAREALIKE(+j) models as the least-supported models. Considering these recent paleobiogeographic studies also pertain to sauropods (generally), it should be worth noting the differences among these studies. This isn't too surprising as the focus of the current study is of a time when the landmasses were coherent and a sympatric model may, in fact, be the likeliest explanation for the pattern whereas the DEC/DIVALIKE(+j) models (vicariance-friendly, if you will) were supported in Gorscak and O'Connor study that focused on a time when the landmasses were breaking apart from one another."*

As is clear from the above, the reviewer actually provides one possible explanation for the differences between our results and those of Gorscak and O'Connor (2016). Given that the latter study dealt with a rather different dataset that pertains largely to the Cretaceous rather than the Jurassic, it is not surprising that it produces a different result. We would perhaps expect a dataset dominated by Cretaceous taxa to show evidence of vicariance and therefore support a model of biogeographic evolution such as DEC/DIVA. Our dataset, however, focuses more on Jurassic taxa, when continental areas had greater connectivity. We might predict therefore that our favoured biogeographic model would be one dominated by processes such as sympatry and dispersal, such as BAYAREALiKE+J. Second, our study differs from that of Gorscak and O'Connor in a very important way. Unlike the latter study, our analyses include information on the paleogeographical relationships between areas through time. The addition of such information should improve the accuracy of biogeographic analyses, and we therefore consider our conclusions to be more likely to be correct than those of Gorscak and O'Connor. Experimentation with different parameters (i.e., running analyses with and without paleogeographical information) indicates that this is the main cause of the differences between our analyses and those of Gorscak and O'Connor. While these results are interesting, we feel that there is no need to add a discussion of this aspect to our paper. This is for two reasons: 1) in order to deal with this topic properly it would require an extensive discussion, which would represent a major aside relative to the main thrust of our paper; and 2) we have already discussed these methodological issues at length previously (Poropat et al. 2016).

4. *"I think the authors should try to implement the new taxon into a recent diplodocoid-focused phylogenetic matrix too (this could be saved for the SI as I doubt it will drastically change their interpretations for it being a diplodocoid of sorts). The Rauhut matrix is great for a sauropod - wide sampling and fits the narrative for pushing neosauropod origins earlier. However, I am curious if these results will hold true in a more specific data set on diplodocoids... if Lingwulong will still hold as a dicraeosaur or potentially as some stem lineage around the base of diplodocoids."*

Done. We have added *Lingwulong* to the data matrix of Tschopp and Mateus (2017), which is the largest and most up-to-date diplodocoid-focused analysis currently available. Full details are provided in the Supplementary Material. In this analysis, *Lingwulong* is placed in essentially the same phylogenetic relationships as we found when using the Rauhut et al. data set. Therefore, our conclusions are strengthened by these results.

5. *“There are many typos and inconsistent formatting in the references for both the main document and the SI. Please fix accordingly.”*

Done. We have corrected all of the typographical and formatting errors in the references noted by the reviewer. We thank the reviewer for spotting these errors.

REVIEWERS' COMMENTS:

Reviewer #2 (Remarks to the Author):

I am satisfied that the authors incorporated the changes suggested from the previous round of reviews—I feel the paper strengthened and more well-rounded. I have no further edits or suggestions to the manuscript and I look forward to seeing this important study published.